# Renin–Angiotensin–Aldosterone System: From History to Practice of a Secular Topic

**DOI:** 10.3390/ijms25074035

**Published:** 2024-04-04

**Authors:** Sara H. Ksiazek, Lilio Hu, Sebastiano Andò, Markus Pirklbauer, Marcus D. Säemann, Chiara Ruotolo, Gianluigi Zaza, Gaetano La Manna, Luca De Nicola, Gert Mayer, Michele Provenzano

**Affiliations:** 16th Medical Department of Internal Medicine with Nephrology & Dialysis, Clinic Ottakring, 1160 Vienna, Austria; sara.ksiazek@hotmail.com (S.H.K.); saemannmarcus@gmail.com (M.D.S.); 2Department of Medical and Surgical Sciences, Alma Mater Studiorum, University of Bologna, 40138 Bologna, Italy; lilio.hu@studio.unibo.it (L.H.); gaetano.lamanna@aosp.bo.it (G.L.M.); 3Nephrology, Dialysis and Renal Transplant Unit, IRCCS Azienza Ospedaliero, Universitaria di Bologna, 40138 Bologna, Italy; 4Department of Pharmacy, Health and Nutritional Sciences, University of Calabria, 87036 Rende, Italy; sebastiano.ando@unical.it (S.A.); gianluigi.zaza@unical.it (G.Z.); 5Centro Sanitario, Via P. Bucci, University of Calabria, 87036 Rende, Italy; 6Internal Medicine IV, Medical University Innsbruck, 6020 Innsbruck, Austria; markus.pirklbauer@i-med.ac.at (M.P.); gert.mayer@i-med.ac.at (G.M.); 7Division of Nephrology, University of Campania “Luigi Vanvitelli”, 80138 Naples, Italy; chiara.ruotolo@yahoo.it (C.R.); luca.denicola@unicampania.it (L.D.N.)

**Keywords:** CKD, ESKD, RAAS, aldosterone, albuminuria

## Abstract

Renin–angiotensin–aldosterone system (RAAS) inhibitors are standard care in patients with hypertension, heart failure or chronic kidney disease (CKD). Although we have studied the RAAS for decades, there are still circumstances that remain unclear. In this review, we describe the evolution of the RAAS and pose the question of whether this survival trait is still necessary to humankind in the present age. We elucidate the benefits on cardiovascular health and kidney disease of RAAS inhibition and present promising novel medications. Furthermore, we address why more studies are needed to establish a new standard of care away from generally prescribing ACEi or ARB toward an improved approach to combine drugs tailored to the needs of individual patients.

## 1. Introduction

The renin–angiotensin–aldosterone system (RAAS) is a complex network designed to ensure the perfusion of vital organs. It connects multiple organ systems and has evolved to regulate blood pressure by volume expansion, sodium and water retention, arteriolar constriction as well as thirst stimulation. The key contributors are hereby stated in order of course of action in the name RAAS itself: (1) renin—produced by the kidneys, (2) angiotensin I and II—activation of which starts from liver angiotensinogen through renin and angiotensin-converting enzyme (ACE), respectively, and binding to angiotensin receptors as well as (3) aldosterone—stimulated by angiotensin and secreted from the zona glomerulosa of the adrenal cortex [1]. The dysregulation of the RAAS system in humans leads to damage and fibrosis of several organs such as the kidney, heart as well as the vascular wall [2,3]. In recent decades, the dysregulation of this mechanism has been considered as pivotal in determining the higher cardiovascular (CV) risk in patients with hypertension, diabetes and cardiovascular disease. Hence, most of the attention from stakeholders and clinicians has focused on limiting the RAAS activity to reduce CV risk in high-risk patients. Intriguingly, time and efforts revealed that RAAS inhibition confers protection against another fearsome outcome, namely the progression of CKD toward kidney failure (KF) [4]. Kidney failure implies a major change in quality of life, exposing patients and nephrologists to planning renal replacement therapy as well as exposing patients to a very poor prognosis regardless of the treatment. The already developed RAAS inhibitors (RAASis), namely angiotensin-converting enzyme inhibitors (ACEis) and angiotensin II type 1 receptor blockers (ARBs) were shown to delay CKD. For such reasons, the nephrology community prompted their prescription and RAASis are now used in most chronic nephropathies [5]. Thus, notwithstanding, a number of patients do not benefit from ACEis and ARBs and, in some cases, despite a good response, patients can face side effects (hyperkalemia and acute kidney injury). Therefore, dose maximization, a main step to ensure nephroprotection, has been difficult to achieve. To improve CKD patients’ prognosis, novel drugs have been discovered and most of them implemented in clinical practice, such as SGLT2 inhibitors (SGLT2-Is) as well as non-steroidal mineralocorticoid receptor antagonists (MRAs) [6,7,8]. These drugs, on top of RAAS inhibitors, confer additional kidney and cardiovascular protection in CKD patients, though not all patients show a proper response. Tailoring treatment to the patient’s individual phenotype and kidney disease cause represents the real future of nephrology. Meanwhile, research has provided us with novel RAAS-inhibiting drug classes [9]. In this review, we discuss the RAAS inhibition system, how it originated and how the novel drugs interfere with the system.

## 2. History and Evolution—400 Million Years of RAAS

When studying the physiology and mechanisms behind the RAAS as well as its effect on the organism, it becomes clear that the RAAS is needed for the survival of humans. By amending hypovolemia or hypotension, it upholds glomerular filtration as well as blood flow through other organs and keeps us alive. This system therefore allows an advantage for survival and reproduction, which makes it a trait that can be passed onto new generations which, in other terms, is a matter of evolution. Natural selection describes the mechanism of evolution wherein organisms that match their environment best are more likely to survive and reproduce; hence, evolution strives for optimization in the reproductive age [10]. In this section, we want to discuss how the RAAS was discovered and give an overview of the current evolutionary theories of RAAS formation and its consequence for humans. 

The research history of the RAAS starts in 1898 with Robert Tigerstedt and Per Bergman, who managed to demonstrate that an extract from rabbit kidney increases blood pressure, which they named renin for its origin. It took another 40 years for the connection between the kidney and blood pressure to be re-discovered by Goldblatt and then another 20 years to identify angiotensin II as an effector of the RAAS [11].

Shortly after the development of the first ACEi, research on the evolutionary history of the RAAS started. To further validate the theory that the RAAS has been handed down through natural selection, studies have been conducted on more primitive species, the first one being Taylor in 1977. While having found renin-like material in vertebrates but none in cartilage fish, he inferred that renin must have first appeared in bony fish, amphibians and subsequently vertebrates [12].

Fournier et al. brought those investigations into the 21st century by conducting phylogenetic analyses of the RAAS’s main related genes in several species: chordates and invertebrates. Their results confirm that the RAAS was formed in the Paleozoic about 400 million years ago. It seems that the RAAS helped to adapt to the changes in environments like salt water and fresh water or in moving from water to land [13].

In conclusion, the RAAS is an immensely sophisticated system that evolution has advanced and refined for over 400 million years in order to keep us alive in dramatically changing environments. When species moved to land, the RAAS may have given a natural selection advantage by retaining sodium in hot and dry areas of the world. As we know, angiotensin II can directly cause sodium retention in the proximal tubules as well as through the secretion of aldosterone, which also leads to an increase in sodium retention in the distal nephron. As humans first developed in the African savannah, sodium was a scarce nutrient, so the RAAS was essential for survival [14,15]. Later on, with the diaspora of humankind as well as the discovery of salt for food preservation or its properties to enhance the taste of dishes, the scarceness of sodium disappeared. On the contrary, we nowadays eat a diet very high in salt [16]. This high-salt consumption contributes to the global burden of hypertension [17,18]. It is hypothesized that hypertension is a disease of modern civilization as it mainly derives from the mismatch between ancestral genes and our current environment [15]. There is also evidence that we have started to adapt to the present climate and the environment we live in, as in northern latitudes alleles of the AGT gene associated with lesser activity of the RAAS are more frequent, whereas African populations carry a promoter variant of AGT associated with higher angiotensinogen levels and increased risk of hypertension [19]. 

Following this hypothesis, it would mean that the ancestral trait of sodium-conserving is maladaptive in the present environment, finally leading to hypertension and increased cardiovascular risk. However, it can also be looked at from another standpoint, where hypertension is more than a bystander effect and the actual trait is deemed more important by evolution. An example of this would be the mutation in the APOL1-gene found more frequently in people of African ancestry: this gene variant is associated with higher risk of renal or cardiovascular disease but also provides protection against Trypanosoma brucei, the cause of sleeping sickness, and is therefore a positive selection advantage [15]. 

Regardless the correct hypothesis, we are left with the question of whether the RAAS is still necessary or maybe even unfavorable to us [20]. From large-scale studies of various medications blocking the RAAS conducted in the last 30 years, we know that inhibition leads to a significant decline in mortality [21,22]. However, studies have also found that too much RAAS blockade, as when using a combination of angiotensin-converting enzyme inhibitors (ACEis) and angiotensin II type 1 receptor blockers (ARBs) together, overcomes the beneficial effects of RAAS inhibition (RAASi), leading to negative effects [23]. There is also a dose-dependent effect of RAASi, where reaching the maximum dose is crucial for reducing proteinuria while the effect on blood pressure may not further increase with maximum doses [24]. 

Currently new drugs targeting the RAAS are constantly being researched in order to optimize RAASi and decrease mortality in patients with hypertension, heart failure, kidney failure or diabetes. As research shows that RAAS inhibition is beneficial but that, on the other hand, a complete inhibition of the RAAS may be harmful, it remains controversial whether the RAAS is still an essential mechanism of evolution or maybe already redundant to humankind [25]. 

## 3. Pharmacological Strategies to Block RAAS and Improve Prognosis in CKD

During the last three decades, RAAS inhibition has revolutionized the therapies to halt the progression of kidney disease and reduce cardiovascular mortality. To acknowledge the role of the RAAS-blocking therapies as standard of care, understanding the physiology of the RAAS is the first step. The hormonal cascade of the RAAS starts with the synthesis of renin from the juxtaglomerular cells around the afferent arteriole; renin secretion is regulated by numerous factors in response to variations in perfusion pressure, changes in NaCl excretion at the level of the macula densa, negative feedback from angiotensin II and via beta-1 adrenergic receptors. Renin degrades angiotensinogen, which is constitutively secreted by the liver, into the peptide angiotensin I (Ang I), which in turn is hydrolyzed by the ACE to the biologically active angiotensin II (Ang II). Ang II causes the constriction of arterial blood vessels, with a consequent rise in blood pressure, and also stimulates the secretion of aldosterone from the adrenal cortex. Ang II also increases sodium reabsorption per se. In turn, aldosterone increases the volume of extracellular body fluid and blood pressure through the reabsorption of sodium, chloride and water from kidney tubules associated with the excretion of potassium and hydrogen ions [26]. The mechanisms of action of the milestone RAAS-blocking drugs are focused on (1) the inhibition of ACE, (2) the blockade of the angiotensin-II-receptor and (3) the antagonism of aldosterone receptors. Notably, ACEis competitively block the action of ACE, hindering the conversion of angiotensin I into angiotensin II, and therefore the secretion of aldosterone. In the 90s, two trials evaluating the ACE inhibitor captopril in patients with CKD and diabetes demonstrated beneficial effects on slowing kidney function decline and progression of proteinuria [27,28]. The rate of increase in serum creatinine was 0.2 ± 0.8 mg/dL/year in patients taking captopril, compared to the rate of 0.5 ± 0.8 mg/dL/year in the placebo group, while concurrently the albumin excretion rate decreased by 17.9% with captopril versus an annual increase rate of 11.8% with placebo. Later in 2000, the HOPE (Heart Outcomes Prevention Evaluation) study over more than 3500 diabetic patients demonstrated the protective properties of ramipril against cardiovascular events and overt nephropathy independent from blood pressure [29]. Subsequently, in 2001 the RENAAL (Reduction of Endpoints in NIDDM with the Angiotensin II Antagonist Losartan) study demonstrated the efficacy of the ARB losartan in diabetic nephropathy to significantly reduce proteinuria by 35% and risk of doubling of serum creatinine level and ESKD by 25% and 28%, respectively [30]. Similarly, the IDNT (Irbesartan Diabetic Nephropathy Trial) showed that irbesartan reduced the risk of proteinuria and doubling of creatinine concentration by 33% and the risk of ESKD by 23% compared to placebo [31]. Moreover, both losartan and irbesartan showed cardiovascular (CV) benefits, lowering the rate of death from CV causes, congestive heart failure, myocardial infarction and cerebrovascular events. Still, despite the general positive results, there is an individual variability in drug response and a residual renal and CV risk remained consistent in up to 40% patients, leading to numerous attempts made over the years to further lower this risk also using the dual blockade of RAAS. The VA-NEPHRON-D trial investigated the combination of ACEi + ARB; however, it resulted in increased risk of adverse events, in particular hyperkalemia and acute kidney injury [32]. The combination of direct renin inhibition with aliskiren and standard of care with ACEi/ARB was analyzed by the ALTITUDE trial; however, the trial was terminated prematurely due to a higher occurrence of primary end-points (composite of the time to CV and renal events) in the aliskiren group [33]. In an interesting analysis of the causes of failure of such large randomized controlled trials, de Zeeuw and Heerspink remarked the need for a future personalized therapy and a more careful selection of the patients before starting the trials, in order to avoid adverse events in patients known to be at risk of complications and who show no initial response of albuminuria lowering to the tested drug [4]. Table 1 summarizes the mechanism of action and prognostic role on CV mortality and ESKD of ACE inhibitors and ARB.

## 4. Novel Drugs That Intervene on the RAAS

Despite the demonstrated renal and CV benefits of standard treatment with ACEis and ARBs, a significant risk in overall mortality still persists [34]. In particular, patients with CKD have a much higher cardiovascular risk when compared to non-CKD patients associated with an increased severity of kidney disease [35]. Apart from this, the accumulation of uremic toxins generates oxidative stress, inflammation and platelets activation, further driving cardiovascular disease [35]. The need for additional risk reduction has led to the search for medications with similar benefits. Many of these effective medications target some parts of the RAAS and researchers have dissected each step of the RAAS pathway in order to find a novel treatment. Apart from new targeting points, already established objectives have been analyzed for an even better aim and treatment effect. In this section, we will describe novel RAASis and their present trial results (see Table 2 and Figure 1). We start with discussing ARNI and nsMRAs who are already being used widely and then move on to describe novel medications while presenting their current status in clinical research. 

### 4.1. Angiotensin Receptor–Neprilysin Inhibitors

Following the success of ACEis and ARBs, a new class of RAASis was introduced in recent years: the angiotensin receptor–neprilysin inhibitors (ARNIs). This class combines an ARB with a neutral endopeptidase inhibitor (NEPi) and the combination of valsartan and sacubitril was the first-in-class ARNI. Natriuretic peptides (NPs) like atrial-NP, B-type NP or c-type NP are normally degraded by neprilysin. NEPis therefore increase the levels of these natriuretic peptides, promoting natriuresis and leading to improvement in myocardial relaxation, reduction in hypertrophy as well as blood vessel dilatation [36]. The two landmark trials PARADIGM-HF and PARAGON-HF demonstrated that ARNIs decrease CV death and heart failure hospitalization by 25–30% as compared to monotherapy with either ACEis or ARBs [35]. Therefore, the indication to use ARNI in patients with HF is justified; nevertheless, the use of ARNI over ACEis/ARBs in CKD patients still remains controversial [35]. In PARADIGM-HF, the annual decline in eGFR was smaller in sacubitril/valsartan compared to enalapril (−1.61 [95%CI: −1.77 to −1.44] vs. −2.04 [95%CI: −2.21 to −1.88]) and also associated with increased albuminuria (1.20 mg/mmol [95%CI: 1.04–1.36] vs. 0.90 mg/mmol [95%CI: 0.77–1.03]) [47]. In PARAGON-HF, sacubitril/valsartan showed a 50% risk reduction in composite kidney events (HR 0.50 [95%CI: 0.33–0.77]), though with no difference in the risk for progression to ESRD [37]. Lastly, the UK-HARP III trial, where 414 CKD patients with an eGFR 20–60 mL/min/1.73 m^2^ were randomized to either ARNI or ARB monotherapy, showed no difference between the two study groups in terms of kidney function or albuminuria [38]. As of now, ARNIs do not seem to have an additional effect on CKD when compared with ACEi/ARB while they should be considered in CKD patients with HF under strict monitoring for hypotension and hyperkalemia [35].

### 4.2. Non-Steroidal Mineralocorticoid Receptor Antagonists

This sub-class of MRAs shows a more selective binding to MR compared to steroidal MRAs. Despite binding to the same ligand-binding domain of the MR, non-steroidal MRAs are characterized by a bulky binding to MR. This causes a conformational modification of the receptor that affects the recruitment of transcriptional coactivators, leading to a lessened transcriptional activation of proinflammatory molecules [48]. Moreover, they show a more favorable risk–benefit profile, shorter half-life and less adverse effects (such as hyperkalemia, gynecomastia, impotence) [48]. Currently, two non-steroidal MRAs have been authorized for clinical use: finerenone and esaxerenone [49]. 

Since the results of the two large phase 3 trials FIDELIO-DKD and FIGARO-DKD, finerenone has changed the landscape of nephrology, cardiology and endocrinology and has already been included into the international KDIGO guidelines for diabetic kidney disease [50]. FIDELIO-DKD primarily looked at kidney failure endpoints, and FIGARO-DKD looked at a cardiovascular composite primary endpoint [7,8]. In either study, finerenone significantly reduced their respective primary endpoints and the results were confirmed in the pooled analysis FIDELITY, including a total of 13,171 patients [41]. Finerenone reduced the composite cardiovascular endpoint of time to cardiovascular death, nonfatal MI, nonfatal stroke or HF hospitalization (HR 0.86, 95%CI 0.78–0.95) and the composite kidney endpoint of time to first onset of kidney failure, sustained eGFR decrease ≥ 57% or renal death (HR 0.77, 95%CI 0.67–0.88) as well as the risk for end-stage kidney disease (HR 0.80, 95%CI 0.64–0.99) [51,52]. It has to be noted that only patients with an initial serum potassium of ≤4.8 mmol/L were included in the trials and the rate of hyperkalemia was 1.7% under active therapy vs. 0.6% in the placebo arm in the pooled FIDELITY analysis [51]. An interesting subgroup analysis of patients receiving SGLT2i in combination with finerenone (6.7% before trial on SGLT2i, 8.5% initiated during trial) showed no interaction between SGLT2i and finerenone in regard to kidney or cardiovascular composite outcomes [41]. It was noted, however, that patients on SGLT2i had a lower incidence of hyperkalemia, suggesting a complementary efficacy of the two drugs [41,52]. A phase 3 trial investigating the efficacy of finerenone in nondiabetic CKD patients is still ongoing (FIND-CKD) [53].

Until now, the non-steroidal MRA esaxerenone has only been approved in Japan for the treatment of hypertension. All phase 3 trials leading up to the approval of esaxerenone were also conducted in Japan: in ESAX-HTN, esaxerenone has shown a dose-dependent blood pressure reduction, which was at least equivalent to eplerenone [54]. In the two other trials conducted in patients with CKD, type 2 diabetes and albuminuria esaxerenone on top of RAASis decreased the urine albumin–creatinine ratio (uACR) by 58% and 54.6%, respectively [42,55]. Hyperkalemia ≥ 5.5 mmol/L was present in 9% and 5.4% of patients and was generally increased within the first two weeks of treatment and more likely occurred in patients with an eGFR < 60 mL/min/1.73 m^2^ [42,49,55]. 

An additional non-steroidal MRA ocedurenone (KBP-5074) is currently being investigated in a phase 3 trial (CLARION-CKD) for efficacy and safety in uncontrolled hypertension and moderate-to-severe CKD [56]. So far, ocedurenone has shown a significant blood pressure reduction of −10.6 mmHg in patients with CKD G3b/4 in its phase 2b study (BLOCK-CKD) when compared with placebo [57,58]. More results from phase 3 trials are awaited in order to fully assess the risk of hyperkalemia as well as to evaluate proteinuria reduction and cardiovascular or mortality outcomes.

### 4.3. Aldosterone Synthase Inhibitors

A different promising mechanism of RAASi is the inhibition of aldosterone synthase (Asi). At variance with MRAs which target to mitigate the effects of aldosterone, Asis act one step earlier by inhibiting the production of aldosterone itself [59]. Baxdrostat is a highly selective Asi, with no impact on cortisol secretion, that has shown promising results in the treatment of resistant hypertension [60]. In a recently published phase 2 trial (BrigHTN), baxdrostat exhibited a dose-dependent reduction in systolic blood pressure when compared with placebo (2 mg vs. placebo: −11.0 mmHg [95%CI −16.4 to −5.5; *p* < 0.001]; 1 mg vs. placebo: −8.1 mmHg [95%CI, −13.5 to −2.8; *p* = 0.003]) [43]. Baxdrostat also seems to have a very safe profile as the investigators deemed there to be few serious adverse events associated with its use [43]. Specifically, only three patients showed serum K ≥ 6 mmol/L, with two of them being able to continue the drug while on dietary potassium restriction [43]. However, it must be emphasized that patients with an eGFR < 45 mL/min/1.73 m^2^ were excluded from the trial. How baxdrostat performs in the currently ongoing phase 3 trial is yet to be presented.

Recently, a new aldosterone synthase inhibitor BI-690517 was presented at the 2023 American Society of Nephology’s (ASN) Kidney Week in Philadelphia; in a phase 2 trial, this new drug showed an uACR reduction of up to 39.5% on top of SGLT2-I therapy [44,61]. 

Notably, the inhibition of aldosterone synthesis, as mentioned, is likely to cause hyperkalemia and hyponatremia, as with MRAs. Moreover, the absence of aldosterone may cause the activation of MR by cortisol, which acts as a mineralocorticoid, resulting in hypertension, sodium retention and hypokalemic metabolic alkalosis [62]. For these reasons, the decrease in aldosterone concentration has been matter of concern and results from clinical trials are eagerly expected.

### 4.4. Aminopeptidase A Inhibitors

A first-in-class aminopeptidase A inhibitor (APAi), firibastat, was tested in a phase 3 trial in patients with therapy-resistant hypertension (FRESH trial) [45]. APA is a central enzyme converting angiotensin II to angiotensin III in the brain. Angiotensin III can increase blood pressure by increasing vasopressin release which inhibits diuresis, stimulating sympathetic tone and vascular resistance as well as stimulating baroreflex function [59]. Therefore, the inhibition of APA is meant to block all these mechanisms and decrease systemic blood pressure. After 12 weeks of treatment with firibastat on top of two to three other blood pressure medications in the FRESH trial, there was no difference between the group receiving firibastat and that receiving placebo (−7.8mmHg vs. −7.9mmHg) [63]. Therefore, the first-in-class APAi failed to show a treatment effect in therapy-resistant hypertension. Further studies on firibastat in different patient populations, including patients with CKD, as well as the efficacy of other APAis are therefore needed.

### 4.5. Angiotensinogen Suppression

A rise in renin after the administration of RAASis is very well documented, as angiotensin II normally suppresses renin via a negative feedback-loop when binding to the angiotensin-1 receptor [64]. Therefore, when RAASis are administered, the suppression of renin is disinhibited, which raises the concern that it may potentially overcome the effect of the RAAS inhibition itself (RAAS-escape phenomenon) [63]. A potential solution to this is provided by a novel RAASi target: the suppression of angiotensinogen (AGT) by using antisense oligonucleotides (ASOs) or small-interference RNA (siRNA) [64]. These RNA-based therapies lead to the degradation of angiotensinogen mRNA, suppress AGT formation and therefore intervene in one of the very first steps of the RAAS pathway. Currently, several phase 1 and 2 studies are being performed and the most promising at the moment is zilebesiran [64,65]. All benefits found with present RAASis in regard to treatment of hypertension, kidney disease, heart failure, atherosclerosis and metabolic disorders are expected to be also present when suppressing AGT [64].

Zilebesiran has shown encouraging results in its phase 1 trial and, more recently, in the phase 2 KARDIA-1 study [46,65,66]. Zilebesiran was tested with different dosages in patients with mild-to-moderate hypertension. Compared with placebo, the mean difference in the 24 h ambulatory systolic blood pressure at 6 months was −11.1 mm Hg with 150 mg, −14.5 mm Hg with 300 mg (6 months), −14.1 mm Hg with 300 mg (3 months) and −14.2 mm Hg with 600 mg. Therefore, zilebesiran not only significantly reduces blood pressure but it also sustains the reduction throughout 6 months without serious adverse events [66]. 

Considering that those RNA-based therapies are administered via quarterly or biannual injections, poor patient compliance may be conquered. However, this long-lasting and efficient inhibition of the entire RAAS cascade remains controversial as most authors see a potential threat in complete RAAS inhibition, mainly related to lack of defense against volume depletion [23,64,67,68]. It remains to be investigated whether the suppression of AGT may also lead to the undesired effects seen in double- or triple-RAASis or if evading the RAAS escape phenomenon leaves us with a safe RAASi.

## 5. Future Perspectives and Conclusions

We outlined the evolution of the renin–angiotensin–aldosterone system to highlight the question of whether the RAAS as a survival trait is still relevant to humankind in the present. The CV risk reduction through RAAS inhibition that has been proven in landmark trials is a significant milestone in the history of nephrology [21,22]. Nonetheless, too much RAAS inhibition by combining two or more RAAS-inhibiting agents seemed to show more harm than benefit as the renal autoregulation collapses under these circumstances [33]. Especially in older patients with CKD, it was shown that, even if treated with a single RAAS inhibitor, they can develop normotensive acute kidney injury [23,59]. It therefore remains to be seen how much RAAS inhibition we should achieve in our patients to obtain the maximum antialbuminuric effect without compromising renal perfusion. 

Moreover, the addition of another RAAS inhibitor in patients who did not respond to a first RAAS inhibitor has shown to fail in warranting a full benefit from this drug class [4]. It seemed similar to prescribing a patient with an infection who does not respond to an antibiotic yet another antibiotic of the same class. Hence, the failure of such “add-on” strategy prompted the development of novel nephroprotective therapies (such as SGLT2 inhibitors and endothelin receptor antagonists) with a different mechanism of action. Large phase 3 clinical trials have shown that SGLT2 inhibitors, when added to RAASis, confer a significant reduction in the risk for CKD progression and CV events [69,70].

A challenge of the future is to understand how to combine these different drugs in the same patient to reach a full renal and CV protection with the lowest rate of side effects. A combination that demonstrated to further reduce CV mortality, CKD progression and undesired patient outcomes is the addition of either SGLT2-Is or MRAs on top of ACEi or ARB therapy [6,7]. Adding either agent to ACEis/ARBs has shown to have an additive effect on uACR reduction [71]. Furthermore, the addition of SGLT2-Is to an RAAS-inhibiting therapy (as well as a combination with MRAs) is useful to counteract the increased risk of hyperkalemia, a dangerous side effect of RAAS inhibition. This is due to the kaliuretic (enhanced elimination of potassium with urine) effect of SGLT2-Is and was shown in FIDELITY and in the ROTATE-3 studies [51,71]. The combination of SGLT2-Is with endothelin receptor antagonists (ERAs) is very promising. In fact, SGLT2-Is have a diuretic effect and balance the sodium (and volume) retention that is caused by ERAs and may contraindicate these drugs in case of already assessed heart failure. Furthermore, the combination of RAASis with ERAs or SGLT2-Is may have synergistic effects by improving anti-fibrotic and anti-proteinuric effects in the kidney through different drug-associated pathways [72]. 

How the already established RAASis will perform with novel RAASis is partially yet to be shown. The Asi BI690517, for example, was studied in patients on top of stable ACEi/ARB therapy and showed a favorable safety profile thus far [73]. Whether a combination of ACEis/ARBs with AGT-suppressing medications, which intervene at a very early stage of the RAAS and have a long lasting effect, is safe, will be studied in the upcoming KARDIA-2 trial [74]. This will hopefully guide us in establishing of how much RAAS inhibition is acceptable.

However, it is important to highlight that the currently used combinations (RAASis + MRAs + SGLT2-Is) still leaves a number of patients at higher risk [75]. Therefore, this suggests that a combination of medications that intervene with the RAAS is possible, but we still need to find the sweet spot of maximum benefit and minimum harm. 

It may also be necessary to individualize treatment strategies. To individualize care is not easy with currently available diagnostic tools. Patients with CKD have multiple mechanisms of damage that may play different roles and have different weights in sustaining kidney damage and establishing future prognosis. Not all mechanisms are active in the same patients and at the same “severity” of disease [5]. Given the heterogeneity of damage, one attempt to overcome this is to identify biomarkers that are the mirror of the “full” individual kidney damage, namely the combination of tubular and glomerular injury [76]. One example is the diabetic kidney disease model for which the combination of multiple serum markers (including MMP-7, MMP-8, MMP-13 and TNFR1) improved the prediction of variability in eGFR decline over time and thus the prognostic uncertainties [77]. One important future step is to better understand the match between the molecular way of action of drugs and the molecular individual pathways of disease and how this correlates with traditional outcome assessment (e.g., albuminuria or eGFR change over time). The new standard of care may not be evident. An individualized combination of drugs driven by the patient’s condition and biomarker panels may be helpful [76]. This novel approach is intriguing but challenging. In clinical practice and research of CKD, personalized nephrology made some steps forwards. It has been shown that not all patients respond with the same magnitude to SGLT2is and MRAs and that some patients who do not respond to SGLT2is will be covered by MRAs (when added to the same patient) and vice versa [71]. This is a significant discovery in nephrology. However, the next step, namely to understand “a priori” which patient characteristics predict the individual response to each drug with reproducibility and accuracy, is still a matter of research and should be proved. In the context of novel RAASis, the accuracy of each of the novel drug in phase 3 trials needs to be proved. Next, these drugs should be tested in combination with SGLT2-Is or ERAs. Finally, and hopefully in cross-over studies, a major question remains to address regarding which are the patients who respond to novel RAASis and not to SGLT2-Is or ERAs or vice versa, those who respond to SGLT2-Is or ERAs but not to novel SGLT2-Is. These answers are eagerly awaited. 

Another important point we learned from the history and pathophysiology of the RAAS is that this system is ubiquitous. The RAAS involves and is active in the brain, in the kidney, in the heart as well as the vessel wall. Hence, many of the roles of the RAAS in physiologic and pathologic conditions are yet to be discovered. One example can be derived from oncology, a field of medicine that guides personalized therapy, the design of novel clinical trials and helps to elucidate the molecular pathways of disease. It has been demonstrated that the ARB losartan is able to relent the growth of a tumor like the glioblastoma via the inhibition of the Ang II/AGTR1 signaling pathway [78]. The Ang II/AGTR1 signaling pathway is per se involved in a complex mechanism that, through the local production of estrogen, promotes the tumor growth and worsens the patient’s prognosis. The complexity and the widespread expression of RAAS in human cells calls for a careful evaluation of its inhibition and the discovery of molecular pathways associated with protection against kidney disease progression. 

We are therefore in need of novel studies on this old system, incorporating the innovative new RAASi drugs where these questions may be answered. On top of that, there is also evidence of a local or tissue RAAS apart from the so-called classical or circulating RAAS. This local RAAS may work independently or together with the circulating RAAS and has both pathogenic and protective pathways. In particular, the latter ones provide even more potential targets for new medications that may stimulate the protective pathways and therefore have an inhibiting effect on the RAAS [79,80]. 

Some limits of the available phase 3 clinical studies on novel RAAS inhibitors need to be pointed out to open possible new perspectives and inspire future investigations. Analysis of the efficacy of finerenone in FIDELIO-DKD and FIGARO-DKD trials was limited to patients with both CKD and type 2 diabetes; therefore, so far, finerenone has been approved to reduce the risk of renal and heart complications in diabetic kidney disease (results from FIND-CKD on nondiabetic patients with CKD are yet to arrive) [7,8,53]. Also, large trials investigating esaxerenone in non-diabetic CKD at different stages are missing, since the ESAX-HTN trial did not include patients with eGFR < 60 mL/min/1.73 m^2^ and ESAX-DN specifically included patients with diabetic nephropathy; at the moment, esaxerenone is approved for the treatment of essential hypertension only [42,54]. Clinical studies on APAis in CKD patients are missing as well, since the first-in-class firibastat was tested in patients with resistant hypertension; however, this was without clinical benefits [63]. The design of trials enrolling patients with different degrees of eGFR and albuminuria will better inform us around this point in the future. 

Therefore, although we already made significant discoveries about this meticulous survival trait we call the RAAS, many questions still remain, and safe and effective standards of care with novel RAAS inhibitors need to be tested.

## Figures and Tables

**Figure 1 ijms-25-04035-f001:**
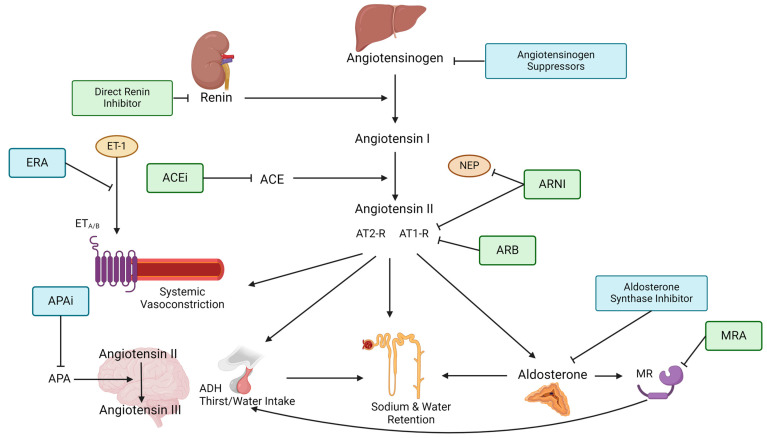
Figure depicts the steps of the RAAS pathway and shows where RAAS-inhibiting therapies target the pathway. The renin–angiotensinogen–aldosterone pathway starts with renin being released from the juxtraglomerullar cells of the kidney due to decreased renal perfusion or low tubular sodium content. Renin then splits angiotensinogen, which is produced in the liver into angiotensin I. Angiotensin I is further converted into angiotensin II by the angiotensin-converting enzyme (ACE). Angiotensin II is the primary effector of the RAAS by causing (1) systemic arteriolar vasoconstriction, (2) secretion of vasopressin (=ADH) in the posterior lobe of the hypothalamus, (3) aldosterone secretion from the zona glomerulosa of the adrenal gland as well as (4) increased sodium and water reabsorption. These effects are mediated through angiotensin II binding to type 1 and type 2 receptors, which often have opposing responses. Aldosterone goes on to bind the mineralocorticoid receptor (MR), causing epithelial sodium channels (EnaC) to be expressed at the apical membrane of the collecting tubules which induces increased sodium and water reabsorption. Aldosterone can also regulate thirst and salt craving by binding to MR in the brain. In the local brain, RAAS angiotensin II is further converted into angiotensin III by aminopeptidase A (APA). Angiotensin III can increase blood pressure by increasing vasopressin release which inhibits diuresis, stimulating sympathetic tone and vascular resistance as well as stimulating baroreflex function. Endothelin-receptor antagonists block the binding of the vasoconstrictive Endothelin-1 to the Endothelin-receptors A and/or B on smooth muscle cells. ACE = Angiotensin-converting enzyme, ACEi = Angiotensin-converting enzyme inhibitor, ADH = Anti-diuretic hormone, APA = Aminopeptidase A, APAi = Aminopeptidase A inhibitor, ARB = Angiotensin II type 1-receptor-blocker, ARNI = Angiotensin II type 1-receptor-neprilysin-inhibitor, AT1-R = Angiotensin type 1-receptor, AT2-R = Angiotensin type 2-receptor, ERA = Endothelin-receptor antagonist, ET-1 = Endothelin-1, ETA/B = Endothelin-receptor type A or B, MR = Mineralocorticoid receptor, MRA = Mineralocorticoid-receptor antagonist, NEP = neutral peptidase.

**Table 1 ijms-25-04035-t001:** Mechanism of action and role on prognosis of ACEis and ARBs.

Class of Drug	Mechanism of Action	Prognostic Role on CV Mortality and ESKD
ACEi	Inhibition of angiotensin-converting enzyme with consequent reduction in angiotensin II formation (vasodilation and decreased blood pressure) and increased bradykinin activity.	- Captopril vs. placebo: 48% risk reduction in doubling of serum creatinine concentration (95%CI 16–69%), 50% risk reduction in the combined end points of death, dialysis and kidney transplantation [27]. - Captopril vs. placebo: reduced progression of microalbuminuria to clinical proteinuria, reduced albumin excretion and preserved creatinine clearance [28].
ARB	Competitive antagonism of angiotensin II receptor with blood pressure-lowering effects.	- Losartan vs. placebo (RENAAL): 25% risk reduction in doubling of serum creatinine concentration (*p* = 0.006), 28% risk reduction in ESKD (*p* = 0.002), 32% risk reduction in first hospitalization for heart failure (*p* = 0.005), average decrease in proteinuria level by 35% (*p* < 0.001) [30]. - Irbesartan vs. amlodipine vs. placebo (IDNT): risk reduction in doubling of serum creatinine concentration by 33% vs. placebo (*p* = 0.003) and by 37% vs. amlodipine (*p* < 0.001); 23% risk reduction in ESKD (*p* = 0.07 for both comparisons) [31].

**Table 2 ijms-25-04035-t002:** Mechanism of action, role on prognosis and differences from ACEi/ARB of the novel RAAS inhibitors drugs.

Class of Drug	Mechanism of Action	Prognostic Role on CV Mortality and ESKD	Difference from ACEi/ARB
ARNI	Block of action of endopeptidase Neprilysin prevents the breakdown of natriuretic peptides BNP and NT-pro BNP, with consequent prolonged duration of natriuretic, diuretic and vasodilation effects. Given that neprilysin breaks down angiotensin II, the effect of accumulation of angiotensin II is blocked by ARB.	- Sacubitril/valsartan vs. enalapril (PARADIGM-HF): slower decrease rate in eGFR (−1.61 mL/min/1.73 m^2^/year vs. −2.04 mL/min/1.73 m^2^/year; *p* < 0.001), greater increase in UACR (1.20 mg/mmol vs. 0.90 mg/mmol; *p* < 0.001) [36]. - Sacubitril/valsartan vs. valsartan (PARAGON-HF): no significant lower rate of total hospitalizations for heart failure (HR 0.85, 95%CI 0.72–1.00) and death for CV causes (HR 0.95, 95%CI 0.79–1.16) [37]. - Sacubitril/valsartan vs. irbesartan (HARP-III): no significant differences in measured GFR and UACR at 12 months; additional reduction in blood pressure and cardiac biomarkers [38].	- The neprilysin inhibitor sacubitril is added to ARB valsartan. - ARNI showed to be superior to ACEi/ARB in reducing the long-term adverse CV outcomes [39] and the risk of all-cause mortality [40].
Nonsteroidal MRA	Decrease in aldosterone effect by binding to the mineralocorticoid receptor, which leads to inhibition of the transcription of specific DNA segments for Na+/K+ ATPase pump at the tubule basolateral membrane and Na+ channel (EnaC) at the apical membrane. It causes increased sodium excretion and higher serum potassium level.	- Finerenone vs. placebo (FIDELIO-DKD): lower risk of CKD progression (HR 0.82, 95%CI 0.73–0.93, *p* = 0.001) and CV events (HR 0.86, 95%CI 0.75–0.99, *p* = 0.03) [7]. - Finerenone vs. placebo (FIGARO-DKD): lower risk of CV events (HR 0.87, 95%CI 0.76–0.98, *p* = 0.03) and renal events (HR 0.87, 95%CI 0.76–1.01) [8]. - Finerenone vs. placebo with and without SGLT2i (FIDELITY): HR for CV events was 0.87 (95%CI 0.79–0.96) without SGLT2i, 0.67 (95%CI 0.42–1-07) with SGLT2i; HR for kidney events was 0.80 (95%CI 0.69–0.92) without SGLT2i, 0.42 (95%CI 0.16–1.08) with SGLT2i [41]. - Esaxerenone vs. placebo (ESAX-DN): higher proportion of patients achieving UACR remission with esaxerenone (absolute difference 18%, 95%CI 12–25%, *p* < 0.001) and higher percent change in UACR from baseline to end of treatment (−58% vs. 8%) [42].	- Inhibition of aldosterone effects. - Finerenone, on top of optimized RAS blockade, provides additional kidney protection compared to optimized RAS blockade monotherapy or dual RAS blockade.
Asi	Selective inhibition of aldosterone synthase (encoded by the gene CYP11B2) with consequent reduction in plasma aldosterone, without affecting 11beta-hydroxylase encoded by CYP11B1 (preserved cortisol synthesis).	- Baxdrostat vs. placebo (BrigHTN): dose-dependent reduction in systolic blood pressure [43]. - BI-690517 on top of SGLT2-I therapy: reduction in uACR of up to 39.5% [44].	- Inhibition of hypersecretion of aldosterone in primary aldosteronism and resistant hypertension.
APAi	Inhibition of the conversion of angiotensin II to brain angiotensin III. It induces improvement in the baroreflex function, decrease in sympathetic tone and vascular resistance, decrease in vasopressin release from the posterior pituitary.	- Firibastat added as a third- or fourth-line therapy (FRESH): no additional drop in systolic blood pressure compared to bi-or triple therapy alone [45].	- Regulation of hypertension by inhibition of brain RAAS.
Angiotensinogen suppressor	Inhibition of hepatic angiotensinogen synthesis through antisense oligonucleotides (ASOs) or small interference RNA (siRNA).	- Zilebesiran vs. placebo (KARDIA-1): improvement in ambulatory and office systolic blood pressure at 3 months and sustained for 6 months (*p* < 0.0001) [46].	- It can circumvent the RAS escape phenomenon (inhibition of negative feedback loop). - Long-term action, potentially requiring 2 injections per year.

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
