# Peer review of "Renin–Angiotensin–Aldosterone System: From History to Practice of a Secular Topic"

_ijms, 2024, doi:10.3390/ijms25074035_

Round 1

Reviewer 1 Report

Comments and Suggestions for Authors

The authors discussed the evolution of the RAAS and its role in human physiology and disease. The RAAS is a complex hormonal cascade that regulates blood pressure, fluid balance, and cardiovascular function. It has evolved over 400 million years, providing a survival advantage in changing environments. However, in modern times, the RAAS may contribute to hypertension and cardiovascular disease due to the mismatch between ancestral genes and the current high-salt diet. The article highlights the benefits of RAAS inhibition in reducing cardiovascular mortality and slowing the progression of CKD.

Comments:

  1. The section on the history and evolution of RAAS is informative but could be more concise. The relevance of some details to the main topic of the review is not always clear.  
  2. The discussion of novel RAAS-targeting drugs would benefit from a more structured approach. Organizing the information into clear subsections based on drug classes or mechanisms of action would improve readability
  3. While the article mentions the potential risks of complete RAAS inhibition, it could provide a more in-depth discussion of the balance between the benefits and risks of RAAS-targeting therapies, particularly in the context of combination therapies
  4. The review does not adequately address conflicting evidence or alternative viewpoints. The
  5. The review does not comprehensively address the limitations of the included studies or the review itself. To improve, the authors should provide a more detailed discussion of the limitations and their potential impact on the conclusions.
  6. The authors mentioned the need for individualized therapy, it does not provide detailed guidance on how to apply the findings in clinical practice. To improve, the authors should offer more specific recommendations for clinicians.
  7. The article briefly mentions the need for further studies but does not provide a comprehensive agenda for future research. To improve, the authors should identify specific knowledge gaps and propose research questions to address them.

Author Response

The authors discussed the evolution of the RAAS and its role in human physiology and disease. The RAAS is a complex hormonal cascade that regulates blood pressure, fluid balance, and cardiovascular function. It has evolved over 400 million years, providing a survival advantage in changing environments. However, in modern times, the RAAS may contribute to hypertension and cardiovascular disease due to the mismatch between ancestral genes and the current high-salt diet. The article highlights the benefits of RAAS inhibition in reducing cardiovascular mortality and slowing the progression of CKD.

R. We would like to thank Reviewer 1 for carefully revising our manuscript. We think that the presentation of the fascinating history of RAAS evolution has some points of interest to introduce the complexity of the contribution of RAAS inhibition to kidney and cardiovascular diseases. In the revised version of our manuscript we have now improved the contents regarding the limitations of RAAS inhibition. We thank for the comments and reported a point-to-point response below.

Comments:

  1. The section on the history and evolution of RAAS is informative but could be more concise. The relevance of some details to the main topic of the review is not always clear.                                                                                    R. Thank you very much. We have made this section more concise to maintain the most relevant concepts. We also improved the quality of writing in the aim of making this part clearer.
  2. The discussion of novel RAAS-targeting drugs would benefit from a more structured approach. Organizing the information into clear subsections based on drug classes or mechanisms of action would improve readability.  R.Thank you. We tried to improve the organization of novel RAAS targeting drugs especially considering the mechanism of action.
  3. While the article mentions the potential risks of complete RAAS inhibition, it could provide a more in-depth discussion of the balance between the benefits and risks of RAAS-targeting therapies, particularly in the context of combination therapies                                                                                            R. We agree. We have provided a deeper discussion about risks and benefits of RAAS-targeting therapies.
  4. The review does not adequately address conflicting evidence or alternative viewpoints. The.                                                                                                      R. Thank you. We added some statements discussing the alternative viewpoints to each drug sections.
  5. The review does not comprehensively address the limitations of the included studies or the review itself. To improve, the authors should provide a more detailed discussion of the limitations and their potential impact on the conclusions.                                                                                                      R. In agreement with the previous comment, we have addressed the limitations of our paper and of the included studies.
  6. The authors mentioned the need for individualized therapy, it does not provide detailed guidance on how to apply the findings in clinical practice. To improve, the authors should offer more specific recommendations for clinicians.                                                                                                                R. The aim of our paper is to illustrate the novel RAS inhibitor therapies with renal protective effects that may have a role beside the standard of therapy. We have now toned-down the concept of individualized therapy because research concerning the specific therapy for personalized medicine is still ongoing and is in the first phases of development. We also explained these points in the “Future perspectives” section by discussing what is the state of art in Personalized medicine in Nephrology and which can be the future steps.
  7. The article briefly mentions the need for further studies but does not provide a comprehensive agenda for future research. To improve, the authors should identify specific knowledge gaps and propose research questions to address them.                                                                                    R. We do agree. We have proposed in the Future Perspectives, some phases for future research that will help to cover the principal questions on novel RAASi but also to understand how to insert these drugs in the scenario of novel therapies with SGLT2i, ERA and MRA. Such inspirations may help to reduce the main gaps in the treatment of kidney diseases, in particular we proposed what can be the future intervention studies around the topic.

Reviewer 2 Report

Comments and Suggestions for Authors

Reviewing the review manuscript entitled, “Renin Angiotensin Aldosterone System: from history to practice of a secular topic” by Ksiazek S H et al., this is an article focusing on history and future related drugs of RAS inhibitors regarding renal protective effects. The authors need to address my concerns below.

 Although the history of RAS in living organisms has been well described, RAS research began with professor Tigerstedt's discovery of renin in 1898. This is an extremely important milestone, and the authors should describe it in the history of RAS.

 ARB indicates that angiotensin II type 1 receptor blocker. The authors should modify it in text.

 The authors need to add AT1 in Figure 1. And the authors should improve to make it easier for readers to distinguish between currently clinically available drugs and investigational drugs.

Author Response

Reviewing the review manuscript entitled, “Renin Angiotensin Aldosterone System: from history to practice of a secular topic” by Ksiazek S H et al., this is an article focusing on history and future related drugs of RAS inhibitors regarding renal protective effects. The authors need to address my concerns below.

R. We would like to thank Reviewer 2 for the appreciation of the idea underlying our manuscript. We have also appreciated their suggestions and comments and following them, we were able to improve the manuscript in terms of readability and completeness of contents. We followed all suggestions and revised the paper, accordingly.

 Although the history of RAS in living organisms has been well described, RAS research began with professor Tigerstedt's discovery of renin in 1898. This is an extremely important milestone, and the authors should describe it in the history of RAS.

R. Thank you for your comment. You will now find a paragraph concerning the research history of the RAAS with the discover of Tigerstedt and Bergman in 1898 mentioned.

 ARB indicates that angiotensin II type 1 receptor blocker. The authors should modify it in text.

R. Thank you for pointing it out. We changed the name ARB to the correct form: angiotensin II type 1 receptor blocker.

 The authors need to add AT1 in Figure 1. And the authors should improve to make it easier for readers to distinguish between currently clinically available drugs and investigational drugs

R. The AT1 and AT2-receptors were also added in our figure and we changed the colours of the novel drugs.